# Field Form of the Dynamics of Classical Many- and Few-Body Systems: From Microscopic Dynamics to Kinetics, Thermodynamics and Synergetics

**Anatoly Yu. Zakharov** 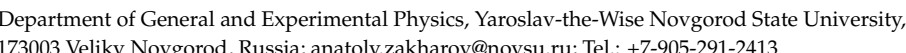

Department of General and Experimental Physics, Yaroslav-the-Wise Novgorod State University, 173003 Veliky Novgorod, Russia; anatoly.zakharov@novsu.ru; Tel.: +7-905-291-2413

**Abstract:** A method is proposed for describing the dynamics of systems of interacting particles in terms of an auxiliary field, which in the static mode is equivalent to given interatomic potentials, and in the dynamic mode is a classical relativistic composite field. It is established that for interatomic potentials, the Fourier transform of which is a rational algebraic function of the wave vector, the auxiliary field is a composition of elementary fields that satisfy the Klein-Gordon equation with complex masses. The interaction between particles carried by the auxiliary field is nonlocal both in space variables and in time. The temporal non-locality is due to the dynamic nature of the auxiliary field and can be described in terms of functional-differential equations of retarded type. Due to the finiteness mass of the auxiliary field, the delay in interactions between particles can be arbitrarily large. A qualitative analysis of the dynamics of few-body and many-body systems with retarded interactions has been carried out, and a non-statistical mechanisms for both the thermodynamic behavior of systems and synergistic effects has been established.

**Keywords:** classical relativistic dynamics; static interatomic potentials; retarded interactions; irreversibility phenomenon; probability-free kinetics; Klein-Gordon equation; principle of causality

## 1. Introduction

Currently, theoretical studies of both thermodynamic properties and kinetic processes of many-body systems are carried out mainly on the basis of statistical mechanics in the framework of the non-relativistic approximation. In this approximation, the interaction between particles is determined by the potential energy, which depends on the instantaneous configuration of the system. As a result, a system consisting of a finite number of particles has a finite number of degrees of freedom. The microscopic dynamics of such a system is described by the deterministic equations of classical mechanics, in which there is no difference between the past and the future. However, such a picture fundamentally contradicts the thermodynamic behavior of systems observed in reality.

A variant of resolving this fundamental contradiction by introducing the concept of probability was proposed by Maxwell [1–3] and Boltzmann [4,5] within the framework of the kinetic theory of gases.

The decisive contribution to the creation of statistical mechanics was made by Gibbs [6], who introduced probability measures in the phase space of many-particle Hamiltonian systems. The construction of the molecular-kinetic theory of Brownian motion by Einstein [7] and Smoluchowski [8] and its triumphant experimental confirmation cast aside "almost" all doubts about the applicability of the concept of probability in physics. However, here it is appropriate to mention the work [9], in which Ritz and Einstein expressed mutually exclusive hypotheses about the nature of the irreversibility phenomenon: "Ritz considers the limitation in the form of retarded potentials as one of the sources of the second law of

thermodynamics, while Einstein suggests that irreversibility based solely on probabilistic grounds". An encyclopedic article by P. Ehrenfest and T. Ehrenfest [10] played an exceptional role in the development of the statistical approach in mechanics. In this work, the methods for calculating probabilities and distribution functions were not so much justified as illustrated using a large number of examples.

Note that the introduction of probabilistic representations into classical dynamics means that the solution of the Cauchy problem for a system of particles is not unique, which contradicts the well-known existence and uniqueness theorem for the Hamiltonian equations of motion of the system. To eliminate this contradiction, it is necessary to introduce an external source that has a non-mechanical nature and affects the dynamics of the system. At the end of the 19th–beginning of the 20th century, there were two mutually irreconcilable concepts containing such a source:

- molecular-kinetic mechanistic theory, in which probabilistic assumptions serve as such a source (Maxwell, Boltzmann, Gibbs);
- the concept of energeticism, in which the very existence of atoms was denied, and the real world is various manifestations of a single hidden substantial and dynamic fundamental principle of the world, called energy (Helm [11], Mach [12], Ostwald [13], Duhem [14]).

Note that both the probability in the kinetic theory and the mysterious unified energy in the concept of energeticism are equally hidden non-mechanical sources, and only the experimental proof of the existence of atoms [15] was the reason for the hasty rejection of the probability-free versions of the microscopic foundation of thermodynamics. Indeed, the existence of atoms does not in any way remove the contradiction between the exact results of classical mechanics (the invariance of dynamics with respect to time reversal $t \rightarrow -t$, the Liouville theorem on the conservation of phase volume, the Poincaré recurrence theorem) and the laws of thermodynamics. Therefore, deterministic classical mechanics and the concept of probability without establishing the physical mechanism of system stochastization mutually exclude each other. In this regard, it is appropriate to note the words of R. Newton [16]: "It should be clear by now that Maxwell's introduction of probabilities had opened a can of worms, but there was no way of getting them back into the can".

The classical notion that interactions between particles can be described in terms of potential energy depending on the instantaneous positions of the particles is limited to the realm of non-relativistic physics. In the framework of the relativistic theory, the interaction between particles is carried out through the field, so the system of interacting particles actually consists of two substances: both particles and the field. Therefore, the dynamics of a system of interacting particles must contain:

1. equations of motion of particles immersed in a field;
2. equations of the dynamics of the field created by these particles.

An example of a theory of this type is classical electrodynamics, in which the interaction between charged particles is carried out through a vector (electromagnetic) field: the field dynamics is described by Maxwell's equations, and the particle dynamics is described by relativistic dynamics [17–19].

The dynamics of a system of particles interacting through a field is fundamentally different from the dynamics of a system of particles with direct instantaneous interactions between them. The reasons for this difference are as follows.

1. Particles and the field are two interconnected subsystems, within each of which there are no interactions. In the general case, a subsystem of a Hamiltonian system is non-Hamiltonian [20]. Although trajectories in the phase space of a subsystem of particles certainly exist, but both the Liouville theorem on the conservation of phase volume and the Poincaré recurrence theorem for a subsystem of particles do not hold.
2. Due to the limited velocity of the field propagation, the instantaneous forces acting on each of the particles of the system are determined by the positions of all other

particles at earlier times. Therefore, the dynamics of the system depends not only on its initial state, but also on its prehistory. Thus, the field character of interactions between particles leads to the phenomenon of heredity.

Starting from 1900 and until recently, several papers have appeared that investigate the dynamics of few-body model systems with signs of thermodynamic behavior. First of all, Lamb proposed a model of an oscillator attached to an infinite string [21,22] and showed that the oscillations of this oscillator are damped. From a modern point of view, the Lamb model is an oscillator immersed in a scalar field with an infinite number of degrees of freedom. The oscillator energy is irreversibly absorbed by this field.

Further, in the papers [23–27], several models of two-body systems with delayed interactions between particles are investigated. The dynamics of such systems is described by functional-differential equations of retarded type. In all the studied models, the irreversibility of the dynamics was established.

Finally, it was established in the paper [28] that the delay in interactions between particles leads to the impossibility of stationary free vibrations of a one-dimensional crystal lattice. Depending on the type of the model potential, only two variants of free vibrations of a one-dimensional lattice are possible.

- Damping of oscillations of all atoms and transition of the system to the state of rest at large times $t \to \infty$. In this case, in the presence of an alternating external field, stationary forced oscillations arise in the system and a dynamic equilibrium is reached between the system of atoms and the external field. In essence, such a state is nothing but a thermodynamic equilibrium between atoms and the field they create.
- The amplitude of at least part of the oscillations increases indefinitely with time. This means the destruction of the lattice.

Within the framework of this model, the relativistic effect of interaction delay is a non-statistical mechanism for establishing dynamic equilibrium in the system "particles + field created by them". This state is identical to thermodynamic equilibrium.

Thus, the dynamics of a classical system of particles within the framework of the field concept of interactions between particles contains the fundamental possibility of describing thermodynamic behavior without using probabilistic assumptions that cannot be verified in any way.

The problem of studying condensed systems within the framework of the non-relativistic physics consists of two parts.

1. Finding interatomic potentials describing the interaction between *resting* atoms. The results of many years of intense efforts to calculate interatomic potentials are systematized in the papers [29–34]. However, the direct use of these results to calculate the thermodynamic and kinetic properties of matter within the framework of statistical mechanics, kinetic theory, and in approaches such as the molecular dynamics method encounters practically insurmountable obstacles. Therefore, in theoretical studies, instead of more or less real interatomic potentials, one has to restrict oneself to simple model potentials, which qualitatively correspond to intuitive physical concepts.
2. Calculation of the partition function of a system of particles interacting through a given interatomic potential. Exact solutions to this problem have been obtained only for the simplest one- and two-dimensional models.

We will assume that the interaction between particles at rest can be represented in terms of the scalar central two-particle potential $v(r)$. This potential will serve as a starting point for the transition from static interatomic potentials to an auxiliary relativistic dynamic field, which is equivalent to interatomic potentials only in the static regime.

## 2. Field-Theoretical Representation of Interatomic Interactions

As is known, interatomic interactions are of electromagnetic origin and only in the case of rest they can be described using instantaneous interatomic potentials. Let us assume that the static scalar interatomic potential $v(r)$ can be represented as a Fourier integral:

$$v(r) = \int \frac{d\mathbf{k}}{(2\pi)^3}\, \tilde{v}(k)\, e^{i\,\mathbf{k}\,\mathbf{r}}, \tag{1}$$

where $r = |\mathbf{r}|$, $k = |\mathbf{k}|$.

*2.1. Rational-Algebraic Model of Interatomic Potentials*

Assume that the function $\tilde{v}(k)$ for real values of $k$ is bounded and is a rational algebraic function of $k^2$.

$$\tilde{v}(k) = \frac{Q_{2m}(k)}{P_{2n}(k)}, \quad (m < n), \tag{2}$$

where $Q_{2m}(k)$ and $P_{2n}(k)$ are polynomials of degree $2m$ and $2n$, respectively:

$$P_{2n}(k) = \sum_{s=0}^{n} C_s\, k^{2s}, \quad Q_{2m}(k) = \sum_{s=0}^{m} D_s\, k^{2s}, \tag{3}$$

$C_s$, $D_s$ are real coefficients.

Since the function $\tilde{v}(\mathbf{k})$ is bounded for all $\mathbf{k}$, it follows that the polynomial $P_{2n}(k)$ has no real roots. We restrict ourselves to the case when the multiplicity of each of the complex roots of this polynomial is equal to one. Then the expansion of the function $\tilde{v}(k)$ into partial fractions has the form

$$\tilde{v}(k) = \sum_{s=1}^{n} \frac{g_s}{k^2 + \mu_s^2}, \tag{4}$$

where $g_s$ and $\mu_s$ are, generally speaking, complex parameters, and $\pm i\mu_s$ are the roots of the polynomial $P_{2n}(k)$.

The function (4) corresponds to the potential of the form

$$v(r) = \frac{1}{4\pi r} \sum_{s=1}^{n} g_s\, e^{-\mu_s r}, \quad \operatorname{Re} \mu_s > 0. \tag{5}$$

The simplest special case, when all $\mu_s$ are real, was studied in [35]. In this case, all the coefficients $g_s$ of the expansion (5) are also real and the corresponding interatomic potentials $v(r)$ can be represented as a linear combination of Yukawa potentials.

Consider the general case when the imaginary parts of at least some of the $\mu_s$ are nonzero

$$\mu_s^{\pm} = \alpha_s \pm i\beta_s, \quad \beta_s \neq 0. \tag{6}$$

Note that the reality of the potential $v(r)$ implies that each pair of mutually conjugate parameters $\mu_s^{+}, \mu_s^{-}$ corresponds to a pair of mutually conjugate parameters $g_s^{+}, g_s^{-}$ that satisfy the condition

$$\operatorname{Im} \left\{ g_s^{+}\, e^{-\mu_s^{+} r} + g_s^{-}\, e^{-\mu_s^{-} r} \right\} = 0. \tag{7}$$

Thus, the total contribution of each pair of mutually complex conjugate parameters $\mu_s^{+}$ and $\mu_s^{-}$ to the total interatomic potential is real and has the form

$$\begin{aligned} v_s(r) &= \frac{1}{4\pi r} e^{-\alpha_s r} (A_s \cos(\beta_s r) + B_s \sin(\beta_s r)) \\ &= \frac{\sqrt{A_s^2 + B_s^2}}{4\pi r} e^{-\alpha_s r}\, \sin(\beta_s r + \psi_s), \end{aligned} \tag{8}$$

where $A_s$ and $B_s$ are real parameters related to $g_s^{\pm}$ by the relation

$$g_s^{\pm} = \frac{1}{2}(A_s \pm iB_s). \tag{9}$$

In this case, at least some of the contributions to the total interatomic potential are oscillating (sinusoidal) potentials whose amplitudes $C_s$ decrease according to the Yukawa law:

$$C_s = \frac{\sqrt{A_s^2 + B_s^2}}{4\pi r} e^{-\alpha_s r}. \tag{10}$$

Thus, the total static interatomic potential $v(r)$, whose Fourier transform $\tilde{v}(k)$ is a rational algebraic function of the square of the wave vector $k^2 = |\mathbf{k}|^2$, can be represented as a linear combination of elementary potentials $v_s(r)$:

$$v_s(r) = \frac{g_s}{4\pi r} e^{-\mu_s r}, \quad \mathrm{Re}\,\mu_s > 0. \tag{11}$$

For $\mathrm{Im}\,\mu_s = 0$ the corresponding elementary potential $v_s(r)$ is a Yukawa potential. For $\mathrm{Im}\,\mu_s \neq 0$ the corresponding contribution to the total interatomic potential consists of pairs of mutually complex conjugate elementary potentials of the form

$$v_s^{\pm}(r) = g_s^{\pm} e^{-(\alpha_s \pm i\beta_s)r}, \quad g_s^{+} = \left(g_s^{-}\right)^{*}. \tag{12}$$

Each of the elementary potentials satisfies the equation

$$\left(\Delta - \mu_s^2\right) v_s(r) = 0. \tag{13}$$

### 2.2. Transition from Interatomic Potentials to Field Equations

In the paper [35] the notion of an auxiliary field $\varphi(\mathbf{r}, t)$ is introduced, which in the static case (i.e., for particles at rest) coincides with the interatomic potential $v(r)$, and in the dynamic case describes the interaction between particles in terms of the classical relativistic field.

The transition from the static field $v(\mathbf{r})$ to the dynamic relativistic field $\varphi(\mathbf{r}, t)$ is carried out in the field equations by replacing the Laplace operator $\Delta$ to the d'Alembert operator $\Box$ [35–37]

$$\Delta \implies \Box = \Delta - \frac{1}{c^2}\frac{\partial^2}{\partial t^2}. \tag{14}$$

Applying this procedure to elementary potentials $v_s(\mathbf{r})$ leads to the Klein-Gordon-Fock equation for elementary auxiliary fields $\varphi_s(\mathbf{r}, t)$

$$\left(\Box - \mu_s^2\right) \varphi_s(\mathbf{r}, t) = 0. \tag{15}$$

Thus, the real auxiliary relativistic field, in terms of which the interaction between particles is described, is a linear combination of, generally speaking, complex elementary fields $\varphi_s(\mathbf{r}, t)$, each of which is characterized by the complex parameter $\mu_s$ and is described by the corresponding Equation (15).

As a result, the system of interacting particles is a union of two subsystems.

1.  Subsystem consisting of particles between which there is no direct interaction. The impact of some particles on others is carried out only through the field created by them.
2.  A subsystem consisting of an auxiliary composite field without direct self-action. The influence of the field at some points on the field at other points is carried out only through particles. Regardless of the number of particles in the system, the auxiliary field has infinitely many degrees of freedom.

### 2.3. Green'S Functions of Elementary Fields and an Abundance of Interaction Retardations

The green function of the Klein-Gordon operator $\hat{L}_s = \Box - \mu_s^2$ is defined by the equation

$$\left(\Box - \mu_s^2\right) G_s\left(\mathbf{r} - \mathbf{r}', t - t'\right) = -\delta\left(\mathbf{r} - \mathbf{r}'\right)\delta\left(t - t'\right) \tag{16}$$

and has the well-known form [38,39]

$$
\begin{aligned}
G_s\left(\mathbf{r} - \mathbf{r}', t - t'\right) &= \frac{\delta\left(t - t' - \frac{|\mathbf{r}-\mathbf{r}'|}{c}\right)}{4\pi|\mathbf{r}-\mathbf{r}'|} \\
&- \theta\left(t - t' - \frac{|\mathbf{r}-\mathbf{r}'|}{c}\right) c\mu_s \frac{J_1\left(\mu_s\sqrt{c^2(t-t')^2 - |\mathbf{r}-\mathbf{r}'|^2}\right)}{4\pi\sqrt{c^2(t-t')^2 - |\mathbf{r}-\mathbf{r}'|^2}},
\end{aligned}
\tag{17}
$$

where $\theta(t)$ is the Heaviside step function, $J_1(x)$ is the Bessel function.

Hence follows the retarded potential of the Klein-Gordon field [39]

$$
\begin{aligned}
\varphi_s(\mathbf{r}, t) = \int d\mathbf{r}' & \left[ \frac{\rho\left(\mathbf{r}', t - \frac{|\mathbf{r}-\mathbf{r}'|}{c}\right)}{4\pi|\mathbf{r}-\mathbf{r}'|} \right. \\
& \left. - \mu_s \int_0^\infty \rho\left(\mathbf{r}', t - \frac{1}{c}\sqrt{\xi^2 + |\mathbf{r}-\mathbf{r}'|^2}\right) \frac{J_1(\mu_s\xi)}{4\pi\sqrt{\xi^2 + |\mathbf{r}-\mathbf{r}'|^2}} \, d\xi \right],
\end{aligned}
\tag{18}
$$

where $\rho(\mathbf{r}, t)$ is the instantaneous microscopic density of the number of particles (atoms):

$$\rho(\mathbf{r}, t) = \sum_a \delta(\mathbf{r} - \mathbf{r}_a(t)). \tag{19}$$

The Formula (18) contains two types of interaction delays between the points $\mathbf{r}$ and $\mathbf{r}'$.

1.  A uniquely defined delay that corresponds to a wave propagating at the speed of light $c$

$$\tau_1 = \frac{|\mathbf{r}-\mathbf{r}'|}{c}. \tag{20}$$

2.  An infinite set of delays

$$\tau_2(\xi) = \frac{\sqrt{\xi^2 + |\mathbf{r}-\mathbf{r}'|^2}}{c} \geq \tau_1, \quad (0 < \xi < \infty), \tag{21}$$

depending on the parameter $\xi$ and corresponding to Klein-Gordon waves propagating with all velocities from 0 up to $c$. Note that the delay $\tau_2(\xi)$ can take on arbitrarily large values, which means that the arbitrarily distant past of the system has a direct influence on its evolution at the current time.

Thus, the connection between the evolution of the relativistic auxiliary field $\varphi(\mathbf{r}, t)$ and the dynamics of the system of particles generating this field is nonlocal both in space variables and in time. Therefore, the interaction between particles carried through the auxiliary field is also nonlocal. Temporal nonlocality is due to the dynamic nature of the auxiliary field and can be described in terms of functional-differential equations of retarded type. It is essential that, according to the formula (21), the delay time of interactions between particles can be arbitrarily large.

Note that the system of particles with delayed interaction is not Hamiltonian. Therefore, many exact results of Hamiltonian mechanics (for example, the Liouville theorem on the conservation of the phase volume, the recurrence Poincaré theorem, etc.), which greatly simplify the qualitative analysis of Hamiltonian systems, do not take place in the dynamics of systems with retarded interactions. Moreover, even the Cauchy problem for

the equations of dynamics of systems with delayed interactions is generally not correct, since the solution of this problem depends not only on the state of the system at the initial moment of time, but also on its entire prehistory (the hereditary effect). In this regard, it is relevant to analyze the qualitative properties of solutions to the equations of dynamics of systems with delayed interactions between particles.

### 3. Qualitative Analysis of System Dynamics within the Framework of the Field Form of Interactions between Particles

*3.1. Two Body Problem*

Consider a model of a system consisting of two particles interacting through the Klein-Gordon field $\varphi(\mathbf{r}, t)$ with parameters

$$\mu^{\pm} = \alpha \pm i\beta. \tag{22}$$

The static potential in this case has the following form

$$v(r) = \frac{A}{4\pi r} e^{-\alpha r} \sin(\beta r + \psi) \tag{23}$$

and has infinitely many minimum points separated from each other by maximum points.

We restrict ourselves to an analysis of the one-dimensional dynamics of this system along the straight line connecting the particles.

In the framework of the non-relativistic theory, each of the minimum points of the potential is a point of stable equilibrium. Near each of the minimums of the potential, the dynamics of the system is close to stationary harmonic oscillations, which can last for an arbitrarily long time.

In the framework of the relativistic theory, there are also infinitely many static equilibrium states in which the distances between particles coincide with the minimum points of the static potential defined by Equation (23). However, as shown in paper [27], in a system of two particles with delayed interactions between them, all equilibrium states are unstable. The fact is that the delay in the interaction between particles leads to the impossibility of stationary harmonic oscillations in the vicinity of a minimum point: infinitely many non-stationary oscillations appear in the system. In this case, the amplitude of at least part of these oscillations increases with time. Thus, the minimum point of the static interparticle potential, which in the framework of non-relativistic dynamics is a point of stable equilibrium, ceases to be such in the framework of the relativistic theory: an arbitrarily small initial perturbation at small times leads to the excitation of multiple harmonics with both increasing and decreasing amplitudes.

The picture of the dynamics of a two-particle system with a multi-well static potential is incomparably more varied than that of a system with one minimum. Let, at the initial moment of time, the system be in the vicinity of some point of minimum of the multi-well static potential. In the vicinity of this point, there are infinitely many non-stationary oscillations with both increasing and decreasing amplitudes. In the case, the system inevitably leaves the vicinity of the initial minimum point and ends up in the vicinity of the neighboring minimum.

Note that the amplitude of spatial oscillations of the static potential in Equation (23)

$$C(r) = \frac{A}{4\pi r} e^{-\alpha r} \tag{24}$$

is a monotonic function of the coordinate $r$, and the distances between the points of neighboring minima of the potential differ little from each other. Therefore, there is a predominant direction of jumps of the system between the points of minima of the static potential $v(r)$: this is the removal of particles from each other, i.e., $r \to \infty$.

However, the situation changes significantly if the total static potential contains the sum of at least two potentials with complex parameters $\mu_1^{\pm}$ and $\mu_2^{\pm}$ ($\mu_1^{\pm} \neq \mu_2^{\pm}$), respectively.

In this case, the distribution of points of minima of the static potential becomes rather irregular, and jumps between neighboring minima become chaotic-like. As an example, Figure 1 shows a qualitative view of a static potential, which is the sum of two elementary potentials with complex parameters $\mu_1^{\pm}$ and $\mu_2^{\pm}$. The set of potential minima in this figure is divided into groups separated from each other by relatively high barriers.

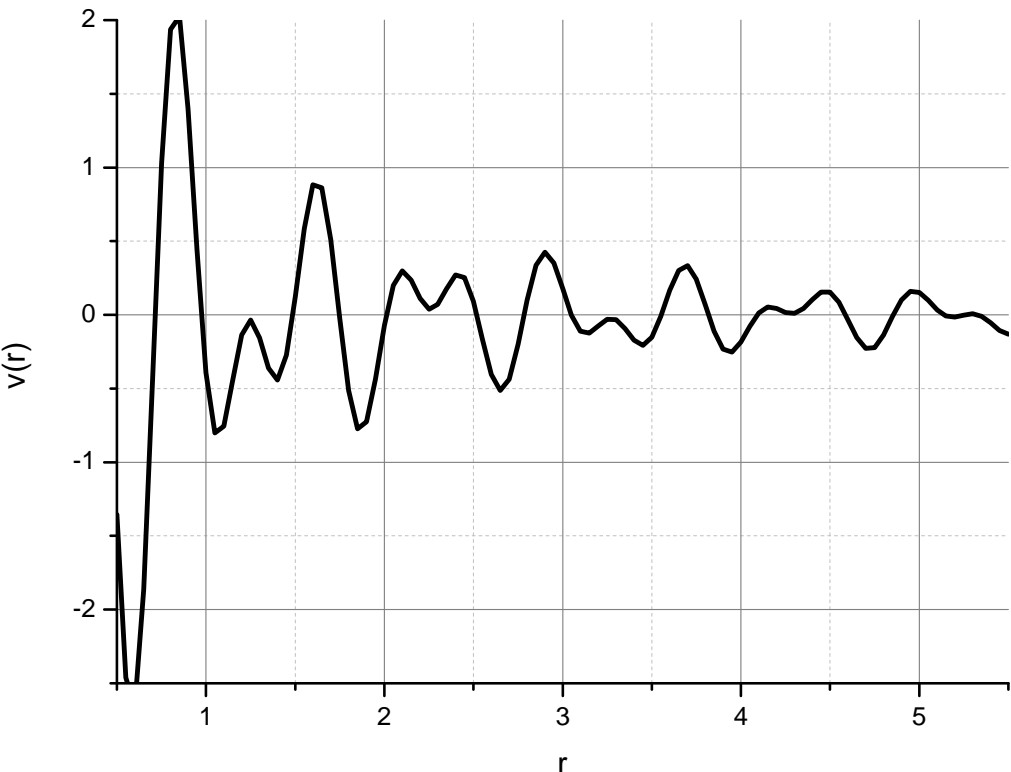

**Figure 1.** Qualitative representation of a static potential, which is the sum of two elementary potentials with complex parameters $\mu_1^{\pm}$ and $\mu_2^{\pm}$.

All jumps of the two-body system between the minima of one group occur more frequently than jumps between different groups. This leads to the appearance of a hierarchy of times in the dynamics of even a two-particle system and has signs of a synergistic effect.

*3.2. Dynamics of a One-Dimensional Crystal and the Establishment of (Thermo) Dynamic Equilibrium*

Similar phenomena take place in the dynamics of the harmonic model of a one-dimensional crystal with retarded interactions between particles [28]. In this crystal model, all frequencies of oscillations are complex, and therefore stationary free oscillations of the system are impossible. Therefore, within the framework of the relativistic dynamics of a harmonic crystal at $t \rightarrow \infty$, only two scenarios of system evolution are possible.

1.  The amplitudes of all free oscillations tend to zero with time. In this case, the energy of the oscillating particles is transferred to the field through which the particles interact. In the absence of a boundary, the field vanishes to infinity, taking energy with it. All free vibrations stop. If the system of particles is placed in a box with impenetrable boundaries for the field, then the field returns to the particles as a force leading to forced stationary oscillations of the particles. This example illustrates a probability-free dynamic mechanism for establishing thermodynamic equilibrium in a system.

2.  Amplitudes of at least part of oscillations of the crystal increase. In this case, the crystal structure is rearranged, the description of which inevitably requires going beyond the limits of the harmonic model. This phenomenon has signs of a synergistic effect.

*3.3. A Rather Amusing Example: Is Confinement Possible in Classical Relativistic Dynamics?*

Note that a function $v_s(r)$ in the formula (11) formally satisfies the Equation (13) not only under the condition Re $\mu_s > 0$, but also under the opposite condition Re $\mu_s < 0$. The second option is usually not considered, assuming that the static inter-particle potential $v_s(r)$ must tend to zero as $r \to \infty$.

Nevertheless, let us consider a static potential of the type (11) for Re $\mu_s < 0$ as applied to the field form of interactions in classical systems

$$v(r) = \frac{C}{4\pi r} e^{\alpha r}, \quad \alpha > 0. \tag{25}$$

This potential tends to infinity both at $r \to +0$ and at $r \to +\infty$, and reaches its minimum value $r = \alpha^{-1}$. Within the framework of classical mechanics, such a potential corresponds to the mutual entrapment of particles and the impossibility of dividing the system of particles into constituent parts. This situation is formally analogous to the phenomenon of quark confinement described in the framework of quantum chromodynamics.

We note the attractive properties of this potential.

- The dynamic field corresponding to this static potential satisfies the Klein-Gordon equation and is therefore relativistic.
- This field is capable of ensuring the stability of a complex consisting of a finite number of particles within the framework of the non-relativistic approximation.

However, the direct use of this potential encounters very significant and yet unsurmounted difficulties, which are as follows.

- When studying the oscillations of a two-particle system in the framework of the relativistic theory, as is known, the complexity of the roots of the characteristic equation leads to the impossibility of stationary oscillations and the loss of stability of the system.
- On the other hand, the infinite distance of particles from each other is hindered by the unlimited growth of the potential at $r \gg \alpha^{-1}$. Unfortunately, a qualitative analysis of the behavior of the system under the condition $r \gtrsim \alpha^{-1}$ encounters obvious fundamental difficulties.

## 4. Discussion and Conclusions

The main principles underlying this work are as follows.

1. A rigorous microscopic substantiation of both thermodynamics and kinetic theory, based only on classical Newtonian mechanics, does not currently exist.
2. Interatomic interactions are of field origin. Therefore, any real system consists of particles and a field generated by these particles and transmitting interactions between these particles.
3. In the case of atoms at rest, the interaction between them can be described by interatomic potentials. But in the case of moving atoms, the interaction is described in terms of an auxiliary scalar relativistic field.
4. The auxiliary scalar field is a superposition of elementary fields, each of which is characterized by its own generally speaking complex mass and satisfies the Klein-Gordon equation. Parameters of elementary fields are uniquely expressed through the characteristics of static interatomic potentials.
5. Due to the finiteness of the masses of elementary fields, the propagation velocity of the Klein-Gordon fields can take on any values that are less than the speed of light. This leads to the fact that the delay of interactions between particles can reach arbitrarily large values.
6. Retardation of interactions between particles is a real physical mechanism leading to the irreversibility of the dynamics of both many-body and few-body systems. Thus, there is no need to use any probabilistic assumptions for the microscopic justification of both thermodynamics and kinetics.

Thus, the following are planned as future areas of research:

- The development of a non-statistical dynamic mechanism of irreversible thermodynamic equilibrium in three-dimensional crystal structures is a generalization of our results [28] obtained for one-dimensional lattices.
- Development of a mathematical apparatus for the theoretical study of the processes of restructuring of the structure of microheterogeneous condensed systems.
- Search for methods for constructing microscopic thermodynamics and kinetics of small systems.

**Funding:** This research received no external funding.

**Data Availability Statement:** Not applicable.

**Acknowledgments:** I am grateful to Ya.I. Granovsky, V.V. Uchaikin, M.A. Zakharov, and V.V. Zubkov for stimulating discussions. My special thanks to the anonymous referees, whose valuable comments helped to improve the work.

**Conflicts of Interest:** The author declares no conflict of interest.

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
