# Peer review of "Field Form of the Dynamics of Classical Many- and Few-Body Systems: From Microscopic Dynamics to Kinetics, Thermodynamics and Synergetics"

_quantumrep, doi:10.3390/quantum4040038_

Round 1

Reviewer 1 Report

In this paper, Zakharov et al. present an interesting expansion by proposing representing the system dynamics with field form for interacting particles. With discreet and elegant elaboration of such idea, this work is already in publishable form, several minor concerns/suggestions are listed below. 1. "...is limited to the realm of non-relativistic physics.In the framework of the ..." Missing a space after "." 2. Eq. 1 and 4 and relevant definitions of v(k). Where k sometimes is bolded vector form while other times is italic modular form. It could cause confusion especially when giving definition of v(k) in equation 4 but using v(k) in the text. If this is on purpose, may the author clarify the idea? 3. "Due to the limited speed of field propagation, the instantaneous forces acting on the particles of the system are determined not only by the instantaneous configuration of the particles of this system, but also by its entire prehistory. Thus, the field nature of interactions between particles leads to the phenomenon of system heredity." As one of the two supporting arguments to differentiate between the system dynamics through field and interactions between particle, which is one essential statement from this work, is there any reference supporting such statement? 4. "We restrict ourselves to an analysis of the one-dimensional dynamics of this system along the straight line connecting the particles. In the framework of the non-relativistic theory, each of the minimum points of the potential is a point of stable equilibrium. In sufficiently small neighborhoods of each of the minima of the potential, the dynamics of the system is "almost" harmonic stationary oscillations." May the authors explain/give reference describing the "almost" harmonic stationary oscillations?

Author Response

Point 1: "...is limited to the realm of non-relativistic physics.In the framework of the ..." Missing a space after "."

Response 1: Thank you for the comment. I inserted the required space.

Point 2: Eq. 1 and 4 and relevant definitions of v(k). Where k sometimes is bolded vector form while other times is italic modular form. It could cause confusion especially when giving definition of v(k) in equation 4 but using v(k) in the text. If this is on purpose, may the author clarify the idea? 

Response 2: Thank you for the comment. You're right. Everywhere in the formulas containing the Fourier transform of the potential, k should be represented in italics. I made the necessary corrections to the manuscript.

Point 3: "Due to the limited speed of field propagation, the instantaneous forces acting on the particles of the system are determined not only by the instantaneous configuration of the particles of this system, but also by its entire prehistory. Thus, the field nature of interactions between particles leads to the phenomenon of system heredity." As one of the two supporting arguments to differentiate between the system dynamics through field and interactions between particle, which is one essential statement from this work, is there any reference supporting such statement?

Response 3: You are right again. I have added the necessary clarifications to the manuscript. Namely: "Due to the limited velocity of the field propagation, the instantaneous forces acting on each of the particles of the system are determined by the positions of all other particles at earlier times. Therefore, the dynamics of the system depends not only on its initial state, but also on its prehistory".

Point 4: "We restrict ourselves to an analysis of the one-dimensional dynamics of this system along the straight line connecting the particles. In the framework of the non-relativistic theory, each of the minimum points of the potential is a point of stable equilibrium. In sufficiently small neighborhoods of each of the minima of the potential, the dynamics of the system is "almost" harmonic stationary oscillations." May the authors explain/give reference describing the "almost" harmonic stationary oscillations?

Response 4: Thank you! Indeed, this paragraph is not well written. Therefore, I replaced the indicated phrase with the following: "Near each of the minimums of the potential, the dynamics of the system is close to stationary harmonic oscillations, which can last for an arbitrarily long time. 

Reviewer 2 Report

In this manuscript, the author presented a quantitive analysis of the dynamics of few-body and many-body systems with retarded interactions.

I found the introduction exciting and well-written, and I especially liked too much the history part of it. 

My only comment to improve the manuscript is to include clear physical examples rather than general terms like interatomic interactions or quark trapping (confinement). Some readers may not know much about these enough to understand the argument. 

Author Response

Point 1: My only comment to improve the manuscript is to include clear physical examples rather than general terms like interatomic interactions or quark trapping (confinement). Some readers may not know much about these enough to understand the argument. 

Response 1:  Thanks for the comment! I have significantly expanded the discussion of the issue of interatomic potentials and added the relevant references in the bibliography [29-34].  As for quark trapping (confinement phenomenon), I added a little discussion of this issue in the vicinity of formula (25). In particular, I noted that "Within the framework of classical mechanics, such a potential corresponds to the mutual entrapment of particles and the impossibility of dividing the system of particles into constituent parts. This situation is formally analogous to the phenomenon of quark confinement described in the framework of quantum chromodynamics."